# *Candida* Genotyping of Blood Culture Isolates from Patients Admitted to 16 Hospitals in Madrid: Genotype Spreading during the COVID-19 Pandemic Driven by Fluconazole-Resistant *C. parapsilosis*

**DOI:** 10.3390/jof8111228

**Published:** 2022-11-21

**Authors:** Judith Díaz-García, Ana Gómez, Marina Machado, Luis Alcalá, Elena Reigadas, Carlos Sánchez-Carrillo, Ana Pérez-Ayala, Elia Gómez-García de la Pedrosa, Fernando González-Romo, María Soledad Cuétara, Coral García-Esteban, Inmaculada Quiles-Melero, Nelly Daniela Zurita, María Muñoz Algarra, María Teresa Durán-Valle, Aída Sánchez-García, Patricia Muñoz, Pilar Escribano, Jesús Guinea

**Affiliations:** 1Clinical Microbiology and Infectious Diseases Department, Hospital General Universitario Gregorio Marañón, 28007 Madrid, Spain; judithdiaz5@gmail.com (J.D.-G.); ana.gomez.nunez@gmail.com (A.G.); marinamachadov@gmail.com (M.M.); luisalcala@efd.net (L.A.); helenrei@hotmail.com (E.R.); cscarrillo@salud.madrid.org (C.S.-C.); pmunoz@hggm.es (P.M.); pilar.escribano.martos@gmail.com (P.E.); 2Instituto de Investigación Sanitaria Gregorio Marañón, 28007 Madrid, Spain; 3CIBER Enfermedades Respiratorias-CIBERES (CB06/06/0058), 28029 Madrid, Spain; 4Clinical Microbiology Department, Hospital Universitario 12 de Octubre, 28041 Madrid, Spain; apayalabalzola@salud.madrid.org; 5Instituto de Investigación Sanitaria del Hospital 12 de Octubre, 28041 Madrid, Spain; 6Clinical Microbiology Department, Hospital Universitario Ramón y Cajal, 28034 Madrid, Spain; mariaelia.gomez@salud.madrid.org; 7Instituto Ramón y Cajal de Investigación Sanitaria (IRYCIS), 28034 Madrid, Spain; 8CIBER de Enfermedades Infecciosas (CIBERINFEC), Instituto de Salud Carlos III, 28029 Madrid, Spain; 9Clinical Microbiology Department, Hospital Universitario Clínico San Carlos, 28040 Madrid, Spain; fgromo@salud.madrid.org; 10Instituto de Investigación Sanitaria del Hospital Clínico San Carlos IdISSC, 28040 Madrid, Spain; 11Clinical Microbiology Department, Hospital Universitario Severo Ochoa, 28914 Leganés, Spain; mcuetara@salud.madrid.org; 12Clinical Microbiology Department, Hospital Universitario de Getafe, 28901 Madrid, Spain; mariacoral.garcia@salud.madrid.org; 13Clinical Microbiology Department, Hospital Universitario La Paz, 28046 Madrid, Spain; mariainmaculada.quiles@salud.madrid.org; 14Clinical Microbiology Department, Hospital Universitario de La Princesa, 28006 Madrid, Spain; nellydaniela.zurita@outlook.es; 15Clinical Microbiology Department, Hospital Universitario Puerta de Hierro Majadahonda, 28220 Madrid, Spain; algarra18@hotmail.com; 16Clinical Microbiology Department, Hospital Universitario de Móstoles, Móstoles, 28935 Madrid, Spain; mduran.hmtl@salud.madrid.org; 17Laboratorio Central de la CAM-URSalud-Hospital Infanta Sofía, San Sebastián de los Reyes, 28703 Madrid, Spain; asanchezg@brsalud.es; 18Medicine Department, Faculty of Medicine, Universidad Complutense de Madrid, 28040 Madrid, Spain

**Keywords:** *Candida*, microsatellite genotyping, blood culture, intra-abdominal samples, COVID-19, candidaemia, Madrid

## Abstract

Background: Candidaemia and invasive candidiasis are typically hospital-acquired. Genotyping isolates from patients admitted to different hospitals may be helpful in tracking clones spreading across hospitals, especially those showing antifungal resistance. Methods: We characterized *Candida* clusters by studying *Candida* isolates (*C. albicans*, *n* = 1041; *C. parapsilosis*, *n* = 354, and *C. tropicalis*, *n* = 125) from blood cultures (53.8%) and intra-abdominal samples (46.2%) collected as part of the CANDIMAD (*Candida* in Madrid) study in Madrid (2019–2021). Species-specific microsatellite markers were used to define the genotypes of *Candida* spp. found in a single patient (singleton) or several patients (cluster) from a single hospital (intra-hospital cluster) or different hospitals (widespread cluster). Results: We found 83 clusters, of which 20 were intra-hospital, 49 were widespread, and 14 were intra-hospital and widespread. Some intra-hospital clusters were first detected before the onset of the COVID-19 pandemic, but the number of clusters increased during the pandemic, especially for *C. parapsilosis*. The proportion of widespread clusters was significantly higher for genotypes found in both compartments than those exclusively found in either the blood cultures or intra-abdominal samples. Most *C. albicans-* and *C. tropicalis*-resistant genotypes were singleton and presented exclusively in either blood cultures or intra-abdominal samples. Fluconazole-resistant *C. parapsilosis* isolates belonged to intra-hospital clusters harboring either the Y132F or G458S ERG11p substitutions; the dominant genotype was also widespread. Conclusions: the number of clusters—and patients involved—increased during the COVID-19 pandemic mainly due to the emergence of fluconazole-resistant *C. parapsilosis* genotypes.

## 1. Introduction

Invasive candidiasis, commonly presented as candidaemia and intra-abdominal infection, is typically hospital-acquired and mostly caused by *Candida albicans*, *Candida parapsilosis*, and *Candida tropicalis* [1,2]. Molecular epidemiology has proven useful in tracking the dynamics of *Candida* spp. infections (e.g., pinpointing the infection source and unraveling outbreaks) [3,4]. Some *Candida* genotypes, commonly referred to as clusters, have been found to cause candidaemia in different patients and could involve patients located in a single hospital ward (intra-ward clusters), patients cared for at a given hospital (intra-hospital clusters), or patients admitted to different hospitals, sometimes located in different cities (widespread clusters) [5,6].

Intra-ward and intra-hospital clusters may suggest active patient-to-patient hospital transmission. In fact, we recently proved that the implantation of a campaign to decrease the number of catheter-related infections correlated with a decrease, not only in the overall number of candidaemia episodes but also in the number of *Candida* clusters [7]. In contrast, widespread clusters involve unrelated patients (not admitted to the same hospital) and may represent genotypes prone to causing candidaemia and not necessarily active hospital transmission [6,8]. Some studies have shown an increase in the number of candidaemia cases during the COVID-19 pandemic, in some cases alongside an increase in patient-to-patient hospital transmission [9,10,11,12,13,14].

We recently conducted a three-year study (*Candida* in Madrid, CANDIMAD) to assess the epidemiology and antifungal susceptibility of a large number of *Candida* spp. isolates from blood cultures and intra-abdominal samples from patients admitted to 16 hospitals located in Madrid, Spain [15,16]. We found that the rate of echinocandin resistance remained low; contrarily, we detected the emergence of fluconazole-resistant *C. parapsilosis* across the region. Fluconazole resistance is a matter of concern in *C. parapsilosis* infections, given the intrinsic lower susceptibility of the species to echinocandins [17,18]. Several questions remained unresolved. First, since the study period (2019 to 2021) spanned a period prior to the COVID-19 pandemic and the pandemic’s first waves, it is unknown whether the number of clusters—and patients involved—rose due to the COVID-19 pandemic in Madrid. Moreover, it is unknown whether highly spread clusters may be found not just in blood cultures but also in intra-abdominal samples. Finally, we studied whether antifungal-resistant genotypes found in the blood may also be found in the intra-abdominal compartment and vice versa.

To elucidate these questions, we genotyped the *C. albicans*, *C. parapsilosis*, and *C. tropicalis* isolates collected in the CANDIMAD study.

## 2. Materials and Methods

### 2.1. Isolates Collected and Studied during the CANDIMAD Study

From 1 January 2019 to 31 December 2021, 1520 *Candida* isolates (*C. albicans*, *n* = 1041; *C. parapsilosis*, *n* = 354, and *C. tropicalis*, *n* = 125) from blood cultures (53.8%, *n* = 817) and intra-abdominal samples (46.2%, *n* = 703) were prospectively collected from 1452 patients cared for at 16 hospitals in Madrid, Spain. Details about the CANDIMAD study and participating hospitals were reported elsewhere [15,16], and the numbers of isolates per participating hospital are shown in Table 1.

Briefly, one incident isolate per species, patient, and compartment (blood culture and/or any intra-abdominal samples) was studied. Most patients (95.7%, *n* = 1389) contributed a single isolate, but 4.3% (*n* = 63) contributed ≥2 isolates.

### 2.2. Microsatellite Genotyping

Species-specific microsatellite markers were used to genotype isolates of *C. albicans* (CDC3, EF3, HIS3 CAI, CAIII, and CAVI) [19,20], *C. parapsilosis* (CP1, CP4a, CP6, and B) [21,22], and *C. tropicalis* (Ctrm1, Ctrm10, Ctrm12, Ctrm21, Ctrm24, and Ctrm28) [23]. Capillary electrophoresis using the ABI 3130xl (Applied Biosystems-Life Technologies Corporation, Carlsbad, CA, USA) analyzer was performed on the PCR products, and electropherograms were analyzed with the GeneMapper v.4.0 software (Applied Biosystems-Life Technologies Corporation, Carlsbad, CA, USA). A molecularly identified control strain from each species was used in each run to ensure size accuracy and avoid run-to-run variations. The allele results were converted to binary data by scoring the presence or absence of each allele. The data were treated as categorical, and the genetic relationship between genotypes was examined by constructing a minimum spanning tree (BioNumerics version 7.6, Applied Maths, Sint-Martens-Latem, Belgium). The isolates were considered to have identical genotypes when they presented the same alleles at all loci. Different genotypes were encoded as follows: CA-X (*C. albicans*), CP-X (*C. parapsilosis*), and CT-X (*C. tropicalis*), X representing the internal code of the genotype in our collection.

Definitions were adopted from a previous study [6]. Briefly, a singleton was defined as a genotype found in a single patient; cluster as a genotype found in samples from ≥2 patients; intra-hospital cluster as involving patients admitted to the same hospital (independently of the time elapsed between the first and last isolate within each cluster); and widespread cluster as involving patients admitted to different hospitals. The genotypes were clonally related when there was only one microsatellite locus difference; a group of clonally related genotypes was referred to as a clonal complex.

We compared proportions using a standard binomial method for the calculation of 95% confidence intervals (Epidat v.4.2 Consellería de Sanidade, Xunta de Galicia, Spain).

## 3. Results

Overall, we detected 1107 genotypes exclusively found in either blood cultures (*n* = 528) or intra-abdominal samples (*n* = 479) or in both compartments (*n* = 100).

### 3.1. Genotypes Found in Blood Cultures and Comparisons with Intra-Abdominal Genotypes

We detected 628 genotypes in the blood, of which 545 (86.8%) were singleton and 83 (13.2%) were clusters. The proportion of singletons versus clusters per species was: 86.7% versus 13.3% (*C. albicans*); 86.8% versus 13.2% (*C. parapsilosis*); and 87.2% versus 12.8% (*C. tropicalis*); no differences reaching statistical significance in the proportion of clusters among species were found (*p* > 0.05) (Table 2).

The 83 clusters involved 272 (33.3%) out of the total number of blood culture isolates (817). The percentage and range of isolates involved in clusters per species were *C. albicans*: 29.2%, 2–15 isolates; *C. parapsilosis*: 41.7%, 2–37 isolates; and *C. tropicalis*: 28.1%, 2–4 isolates. *C. parapsilosis* presented a higher proportion of isolates in clusters (*p* < 0.05; Table 2). The cluster analysis showed that a high proportion was widespread (75.9% widespread versus 24.1% intra-hospital clusters; *p* < 0.05); a number of clusters were both widespread and intra-hospital (Table 2). Figure 1 depicts a timeline for *C. albicans* (*n* = 22), *C. parapsilosis* (*n* = 18), and *C. tropicalis* (*n* = 1) intra-hospital clusters. All clusters may involve patients admitted to the hospital within a limited time period or at distant points in time. Several clusters were present before the onset of the COVID-19 pandemic; however, the number of *C. parapsilosis* clusters (and patients involved) was boosted during the pandemic, especially as of the second wave (Figure 1 and Figure 2).

In fact, the percentage of patients involved in intra-hospital clusters of *C. albicans* (12.8%/7.8%/11.2%), *C. parapsilosis* (10.3%/17.2%/44.3%), and *C. tropicalis* (0%/0%/10.5%) detected in the years 2019/2020/2021, respectively, increased over time; these differences only reached statistical significance for *C. parapsilosis* (2019 versus 2021, and 2020 versus 2021 *p* < 0.05; Figure 2).

Patients involved in *C. albicans* intra-hospital clusters were cared for in intensive care units (ICU) (32%), medical (36%), surgical (12%), oncology–haematology (10%), and neonatology (10%) wards. In contrast, patients with *C. parapsilosis* intra-hospital clusters were more frequently cared for in the ICU (62.9%) than other wards (medical 18.6%; surgical 8.6%; oncology–haematology 8.6%; and others 1.4%). Percentage differences in the patients involved in *C. albicans* and *C. parapsilosis* intra-hospital clusters admitted to ICU wards reached statistical significance (*p* < 0.05). Table 3 indicates the frequencies of widespread clusters.

Overall, hospitals 1 to 5, 7, and 9 contributed the highest numbers of widespread clusters, probably as a consequence of the size of the hospital, given that those hospitals showed high rates of candidaemia incidence and number of admissions. Table 4 shows that candidaemia incidence rates varied among hospitals, species, and over time, and the highest incidence rates were observed in 2020 and 2021 (in the hospitals for which admissions and incidence data were available). *C. albicans* incidence increased between the years 2019 and 2020 in most hospitals (11/16), and the highest rates were usually found in 2020 (10/16 hospitals), especially in hospitals 7 and 9. *C. parapsilosis* incidence increased between the years 2019 and 2020 in 7/16 hospitals. In contrast, overall candidaemia incidence gradually increased over the years in six hospitals and was mainly driven by hospitals 4, 9, and 10, the ones in which fluconazole-resistant *C. parapsilosis* clones spread [18]. No statistically significant differences between years were observed for *C. tropicalis* incidence.

Widespread *C. albicans* clusters involved 2 to 7 (the latter concerning the CA-048 cluster) hospitals each; 12 clonal complexes were found (nos. 2, 3, 4, and 5 were previously detected (6), and nos. 6–12 were newly reported herein). Clonal complex no. 5 involved a large number of clusters, isolates, and hospitals (Figure 3).

Likewise, widespread *C. parapsilosis* clusters involved two to nine (the latter concerning the CP-023 cluster) hospitals; four clonal complexes were found (no. 2 was previously detected [6], and nos. 3 and 4, and were newly reported herein) (Figure 4).

Cluster CP-451 was found in four hospitals and involved a high number of isolates. Finally, widespread *C. tropicalis* clusters involved two to four (the latter concerning the CT-226 cluster) hospitals and were unrelated; clonal complexes were not detected.

We simultaneously collected isolates from blood cultures and intra-abdominal samples in 32 patients. Of these, identical genotypes were found in the following patients: *C. albicans*: 14/19; *C. parapsilosis*: 4/8; and *C. tropicalis*: 4/5. We found 579 genotypes in intra-abdominal samples, and the 100 genotypes also found in blood cultures mainly involved different patients: *C. albicans* (*n* = 59/66), *C. parapsilosis* (*n* = 21/24), and *C. tropicalis* (*n* = 8/10). Interestingly, the proportion of widespread clusters was significantly higher in genotypes found in both compartments than in those exclusively found in either blood cultures or intra-abdominal samples: *C. albicans* (77.3%/5.4%/2.9%), *C. parapsilosis* (83.3%/3.2%/0.0%), and *C. tropicalis* (60.0%/5.4%/4.2%), respectively, (*p* < 0.05) (Figure 3b and 4b).

### 3.2. Genotypes Involving Antifungal-Resistant Isolates

Table 5 summarizes the isolates presenting antifungal resistance to fluconazole or echinocandins.

The frequencies of resistant genotypes exclusively found in blood cultures/intra-abdominal samples/both compartments were *C. albicans*: 4/4/1; *C. parapsilosis*: 2/1/2; and *C. tropicalis*: 1/2/0, respectively. Most *C. albicans-* and *C. tropicalis*-resistant genotypes were present exclusively in one compartment and were, with a few exceptions, singletons. The echinocandin-resistant and intra-hospital CA-0511 cluster was found in blood cultures and intra-abdominal samples from two patients admitted to the ICU of Hospital 6; one of the patients also harboured the clonally related CA-1341 echinocandin-resistant genotype. In addition, we found three widespread clusters (CA-056, CA-313, and CT-031) that involved a resistant isolate and some susceptible isolates each. Cluster CA-056 was found in four hospitals (in blood cultures and intra-abdominal samples); clusters CA-313 (only in blood cultures) and CT-031 (only intra-abdominal samples) were found in two hospitals each (Table 5 and Figure 3).

Data concerning fluconazole-resistant *C. parapsilosis* isolates were reported elsewhere (15). All isolates belonged to either a clonal complex with genotypes harbouring the Y132F ERG11p substitution (CP-451, CP-673, and CP-674) or an unrelated cluster with the G458S ERG11p substitution (CP-675). All were exclusively intra-hospital clusters except for the CP-451 cluster, which was widespread and found in four hospitals (Table 5, Figure 1 and Figure 4). Noteworthy, although genotypes CP-451 and CP-675 were found in both compartments, the vast majority of isolates came from blood cultures. The CP-451 genotype was first detected in intra-abdominal samples before the COVID-19 pandemic onset in Hospital 9, and its spread in blood cultures gained traction during the COVID-19 pandemic. Moreover, the CP-451, CP-673, and CP-674 clusters were mainly responsible for the spread of *C. parapsilosis* clusters during the COVID-19 pandemic, and these genotypes accounted for 63% (*n* = 44/70) of the total number of patients involved in *C. parapsilosis* clusters. In fact, when resistant genotypes were excluded from the analysis, the percentage difference of patients involved in clusters in 2019/2020/2021 did not reach statistical significance (10.3%/17.2%/44.3% including fluconazole-resistant *C. parapsilosis* genotypes versus 7.7%/9.2%/11.3% excluding fluconazole-resistant *C. parapsilosis* genotypes) (Figure 2).

## 4. Discussion

By genotyping *C. albicans*, *C. parapsilosis*, and *C. tropicalis* isolates collected in 16 hospitals located in the Madrid metropolitan area, we demonstrated that the number of clusters—and patients involved—increased during the COVID-19 pandemic and this increase was mainly driven by fluconazole-resistant *C. parapsilosis* genotypes. Highly spread genotypes could be found not only in blood cultures but also in intra-abdominal samples, but antifungal-resistant genotypes were mainly present in blood cultures.

Surveys to study *Candida* spp. isolates from blood cultures and intra-abdominal samples are helpful to monitor species epidemiology and antifungal resistance, especially when genotyping of isolates is performed, and may help track antifungal-resistant clone spreading. Whereas the genotypes found in intra-abdominal samples may cause endogenous infection, those found in blood are susceptible to patient-to-patient transmission via the exogenous route [3]. Previous studies from our group demonstrated the usefulness of *Candida* genotyping to unravel the dynamic of isolate transmission within the hospital, as well as distinguish the presence of clusters that may suggest a common source of infection or patient-to-patient transmission, and cause infections in the form of outbreaks. We studied isolates causing candidaemia in hospital-admitted neonates and proved that some clusters were responsible for candidaemia outbreaks in that setting and that they may involve a large number of patients as far as *C. parapsilosis* clusters were concerned [4]. In another study, we demonstrated that the implementation of measures to decrease catheter-related infections correlated with a decrease in the number of clusters and patients with candidaemia involved, and therefore, genotyping proved the benefits of the implemented measures [7].

The CANDIMAD study is a first-in-its-class study conducted in Madrid and was devoted to assessing the burden of antifungal resistance in the city’s metropolitan area [15,16]. It allowed us to demonstrate that the rate of resistance to echinocandins was not a threat, whereas the rate of fluconazole resistance emerged in *C. parapsilosis* because of a genotype spreading across the region. In the present report, we complemented such observations by adding genotyping data on *C. albicans*, *C. parapsilosis*, and *C. tropicalis* blood cultures and intra-abdominal isolates collected in the CANDIMAD study. We detected 83 clusters in blood cultures. Intra-ward clusters (involving patients admitted to the same hospital ward) are likely indicative of hospital patient-to-patient transmission; however, we were unable to track intra-ward clusters in the present study due to the restructuring of hospitals during the COVID-19 pandemic. Therefore, potential patient-to-patient transmission was measured by the presence of intra-hospital clusters, which represented 34/83 of all clusters. Intra-hospital *C. parapsilosis* clusters involved a higher number of patients than those of *C. albicans*, as previously reported [4]. Of note, many of the clusters were detected in patients before the onset of the COVID-19 pandemic, but the number of patients involved in clusters was boosted (particularly for *C. parapsilosis*) during the pandemic, probably because of worsened catheter care.

The presence of genotypes involving epidemiologically unrelated patients (not admitted to the same hospital) is not infrequent, and its interpretation may be controversial [6]. Widespread clusters involve patients admitted to different hospitals and might indicate genotypes actively transferred among hospitals rather than active patient-to-patient hospital transmission. Alternatively, widespread clusters may comprise genotypes prone to cause invasive candidiasis because they are frequently present in the microbiota of healthy subjects or the environment. A number of the clusters detected in Madrid were widespread (63/83), and, as expected, they were found in the largest hospitals. The transfer of patients from smaller to large hospitals may also explain the presence of widespread clusters. Furthermore, clonal complexes are groups of genetically related widespread clusters [6], and their presence strengthens the hypothesis of frequently found genotypes prone to cause invasive candidiasis. In fact, when intra-abdominal genotypes were taken into account, the number of clonal complexes soared. It is worth noting that some of the clonal complexes found were previously detected in Madrid and outside Spain [6].

We detected an increase in the incidence of candidaemia during 2020 and 2021 in some participating hospitals, which is in line with previous reports conducted during the COVID-19 pandemic [9,10,11,12,13,14]. Such an increase could have been due to either an increase in the number of patients prone to develop candidaemia or higher patient-to-patient transmission. Our study included a large number of isolates collected before and during the COVID-19 pandemic and demonstrated that the main driver of the increase in patients involved in clusters during the COVID-19 pandemic in the Madrid region was the emergence of the widespread CP-451 fluconazole-resistant *C. parapsilosis* cluster harbouring the Y132F ERG11p substitution. This genotype was detected for the first time before the onset of the COVID-19 pandemic in an intra-abdominal sample in Hospital 9, where it caused an outbreak later on; it then became widespread across other hospitals, and its spread was reinforced in 2021. The transfer of patients and/or hospital material during the COVID-19 pandemic across the region may explain such spreading. Other *C. parapsilosis* and *C. albicans* clusters were also widespread but involved isolates to a lesser extent, suggesting a high potential for the dissemination of the CP-451 genotype. On the contrary, *C. albicans* antifungal-resistant genotypes each involved single isolates. Other studies conducted in Greece and Brazil also showed that fluconazole resistance in *C. parapsilosis* increased during the pandemic, probably also as a consequence of clonal spreading that could have taken place during the COVID-19 pandemic [11,14].

Our study was subject to some limitations. We were unable to study environmental isolates collected near patients and from healthcare workers, and therefore, we cannot rule out the environmental niches of the isolates or patient-to-patient transmission. Due to the lack of information regarding the candidaemia source, we were unable to study the potential correlation between catheter-related infection and intra-hospital clusters. Moreover, widespread cluster tracking would have been carried out if information regarding patient or medical material transfers across hospitals had been available. Finally, we could not rule out that some clusters were a consequence of the lack of discrimination in the typing procedure, although previous studies suggested that the microsatellite markers used herein were highly discriminative [3,4].

## 5. Conclusions

In conclusion, the number of clusters—and patients involved—increased during the COVID-19 pandemic mainly due to the emergence of fluconazole-resistant *C. parapsilosis* genotypes, predominantly in blood cultures. Our study sets an example of the value of conducting surveys of antifungal resistance in *Candida* spp. and the role of genotyping to track down genotype spreading across hospitals and warrants the continuation of such initiatives in the region.

## Figures and Tables

**Figure 1 jof-08-01228-f001:**
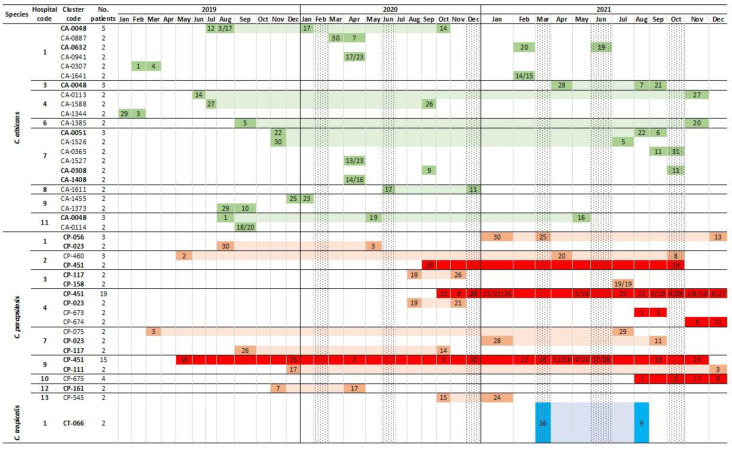
Timeline of *C. albicans* (green shades), *C. parapsilosis* (red shades), and *C. tropicalis* (blue shades) intra-hospital clusters found in blood cultures during the study period (2019–2021). Genotype codes in bold indicate those that were also widespread clusters. Cells in dark colours with numbers indicate the date the genotype was detected in a given patient within each cluster; cells in light shades indicate the latency period (time period spanning two time points in which isolates in a cluster were detected) of the cluster in the hospital. Cells in dark red indicate fluconazole-resistant *C. parapsilosis* genotypes. Dotted cell columns represent the onset of each COVID-19 wave.

**Figure 2 jof-08-01228-f002:**
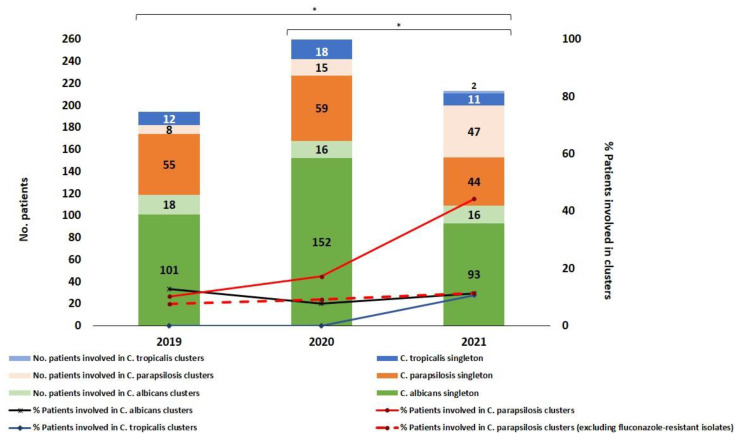
Numbers (and percentage) of patients involved in singleton and intra-hospital clusters per species over the study period. * Differences reaching statistical significance (*p* < 0.05).

**Figure 3 jof-08-01228-f003:**
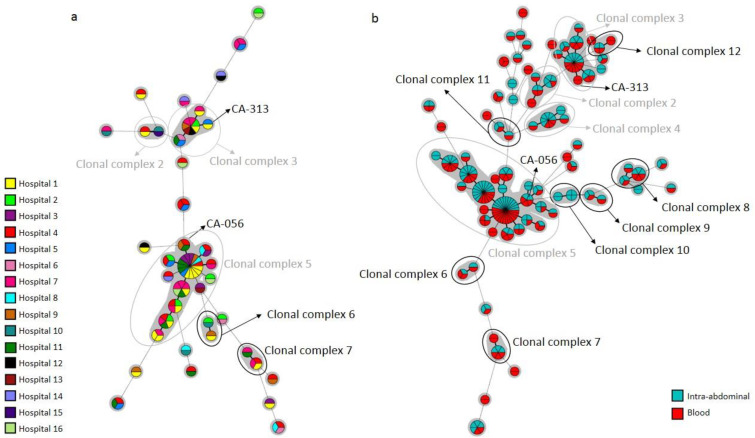
Minimum spanning tree showing widespread *C. albicans* clusters found in blood cultures shown per hospital of source (**a**) or combined with intra-abdominal samples (**b**). Circles represent different genotypes and circle size the number of isolates belonging to the same genotype. Connecting lines between circles show profile similarities. The solid bold line indicates differences in only 1 marker, solid line indicates differences in 2 markers, dashed line indicates differences in 3 markers, and dotted line indicates differences in 4 or more markers. CA-056 and CA-313 were resistant genotypes. Grey circles depict previously reported clonal complexes 2 to 5. Black circles depict clonal complex 6–12, newly reported herein.

**Figure 4 jof-08-01228-f004:**
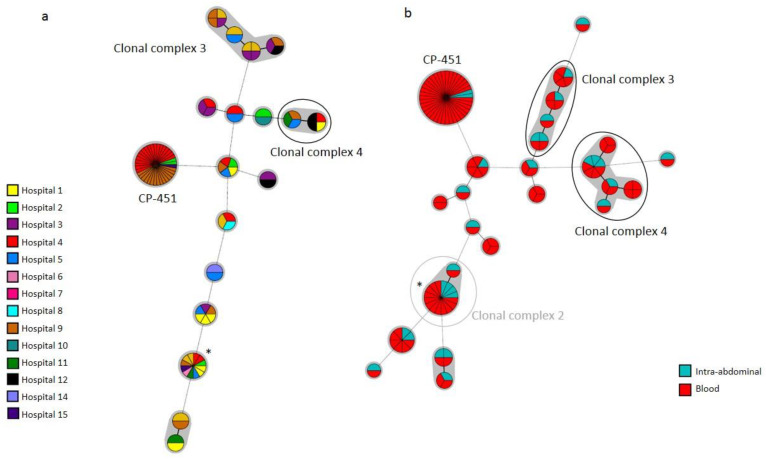
Minimum spanning tree showing *C. parapsilosis* widespread clusters found in blood cultures shown per hospital of source (**a**) or combined with intra-abdominal samples (**b**). Circles represent different genotypes and circle size the number of isolates belonging to the same genotype. Connecting lines between circles show profile similarities. The solid bold line indicates differences in only 1 marker, solid line indicates differences in 2 markers, dashed line indicates differences in 3 markers, and dotted line indicates differences in 4 or more markers. CP-451 is a resistant genotype. The grey circle depicts the previously reported clonal complex 2. Black circles depict clonal complexes 3 and 4, newly reported herein. * Genotype CP-023.

**Table 1 jof-08-01228-t001:** Number of isolates studied (per species and source) collected during the CANDIMAD study at each of the 16 participant hospitals.

Hospital Code	Clinical Source and No. Isolates						
	Blood Cultures			Intra-Abdominal Samples			Overall
	*C. albicans*	*C. parapsilosis*	*C. tropicalis*	*C. albicans*	*C. parapsilosis*	*C. tropicalis*	
1	90	32	16	88	18	12	256
2	34	20	5	104	9	16	188
3	42	22	2	85	9	8	168
4	75	57	6	18	4	1	161
5	30	13	3	44	13	4	107
6	21	4	3	67	4	10	109
7	57	36	8	5	2	0	108
8	25	5	2	57	11	4	104
9	41	35	4	19	5	3	107
10	20	8	2	39	6	4	79
11	21	12	2	6	0	2	43
12	11	9	2	7	0	1	30
13	4	7	1	8	1	1	22
14	6	6	1	2	0	1	16
15	5	5	0	2	1	1	14
16	7	0	0	1	0	0	8
Overall	489	271	57	552	83	68	1520 *

* A total of 62 isolates (4.1%; *C. albicans*, *n* = 10; *C. parapsilosis*, *n* = 49; and *C. tropicalis*, *n* = 3) were found to be antifungal resistant and sourced from either blood cultures (*n* = 51; *C. albicans*, *n* = 5; *C. parapsilosis*, *n* = 45; *C. tropicalis*, *n* = 1) or intra-abdominal samples (*n* = 11; *C. albicans*, *n* = 5; *C. parapsilosis*, *n* = 4; *C. tropicalis*, *n* = 2) [15,16]. Hospital codes were assigned based on the number of isolates collected at each participating hospital, with Hospital 1 being the hospital contributing the highest numbers and Hospital 16 contributing the lowest numbers.

**Table 2 jof-08-01228-t002:** Number of isolates, patients, and genotypes found in blood per species, and genotype analysis.

	*C. albicans*	*C. parapsilosis*	*C. tropicalis*
Isolates/patients	489	271	57
Genotypes	399	182	47
Singleton, n (%)	346 (86.7%)	158 (86.8%)	41 (87.2%)
Cluster, n (%)	53 (13.3%)	24 (13.2%)	6 (12.8%)
Isolates/patients in cluster, n (%)	143 (29.2%)	113 (41.7%)	16 (28.1%)
Range of isolates/patients per cluster	2–15	2–37	2–4
Intra-hospital cluster, n (%)	14 (26.4%)	6 (25.0%)	0 (0.0%)
Widespread cluster, n (%) *	39 (73.6%)	18 (75.0%)	6 (100%)

* A total of 14/63 widespread genotypes were also intra-hospital genotypes (*C. albicans* [*n* = 6], *C. parapsilosis* [*n* = 7], and *C. tropicalis* [*n* = 1]).

**Table 3 jof-08-01228-t003:** Distribution of widespread clusters of *C. albicans* (green shades) and *C. parapsilosis* (pink shades) among the participant hospitals.

Hospital Code	Frequencies of Combinations of Widespread Genotypes																Numbers of Widespread Genotypes		
	1	2	3	4	5	6	7	8	9	10	11	12	13	14	15	16	*C. albicans*	*C. parapsilosis*	Total
** *1* **		3	2	6	2	0	7	1	4	0	3	2	1	0	0	1	15	5	20
** *2* **	2		0	3	1	1	3	0	1	1	1	1	1	0	0	2	8	4	12
** *3* **	1	0		2	1	0	0	2	1	0	1	0	1	0	0	0	4	6	10
** *4* **	3	3	1		4	1	4	3	3	0	5	0	0	1	0	1	17	7	24
** *5* **	3	2	1	3		1	1	1	1	0	3	0	0	0	0	0	7	7	14
** *6* **	1	1	0	1	1		0	1	0	0	1	0	0	0	0	0	3	1	4
** *7* **	1	1	2	2	2	1		0	1	1	3	1	1	1	0	1	12	6	18
** *8* **	0	0	0	1	0	0	1		1	1	1	0	0	0	0	0	4	1	5
** *9* **	3	3	3	3	4	1	3	0		0	2	1	1	0	0	0	6	8	14
** *10* **	0	1	1	0	0	0	2	0	2		0	0	0	0	1	0	4	3	7
** *11* **	2	1	0	1	2	1	1	0	2	0		0	0	0	0	1	8	3	11
** *12* **	1	0	2	1	0	0	0	0	1	0	0		1	1	0	0	3	3	6
** *13* **	0	0	0	0	0	0	0	0	0	0	0	0		0	0	0	2	0	2
** *14* **	0	0	0	0	1	0	0	0	0	0	0	0	0		0	0	3	1	4
** *15* **	1	2	0	2	1	1	1	0	2	0	1	0	0	0		0	1	2	3
** *16* **	0	0	0	0	0	0	0	0	0	0	0	0	0	0	0		4	0	4

**Table 4 jof-08-01228-t004:** Number of admissions and incidence of candidaemia, overall and per species, per participant hospitals, and overall, during 2019, 2020, and 2021.

Hospital Code	Candidaemia Incidence (Cases per 100,000 Hospital Admissions)					
	Overall (No. of Admissions) *			*C. albicans* (2019/2020/2021)	*C. parapsilosis* (2019/2020/2021)	*C. tropicalis* (2019/2020/2021)
	2019	2020	2021			
1 ^α^	114.8 (47,048)	155.5 (42,444)	135.7 (35,377)	53.1/69.6/70.7 ^α^	31.9/16.5/28.3	10.6/14.1/14.1
2	50.7 (45,357)	63.0 (39,669)	78.3 (39,607)	22.0/35.3/25.2	15.4/7.6/25.2	4.4/2.5/5.0
3 ^β^	83.2 (32,442)	51.4 (60,348)	83.3 (40,832)	40.1/24.9/34.3	21.6/13.3/17.1	0.0/0.0/4.9
4 ^γ^	152.9 (31,399)	201.1 (27,841)	231.6 (28,067)	86.0/89.9/81.9	38.2/53.9/106.9 ^β,γ^	12.7/3.6/3.6
5	130.7 (16,064)	208.8 (13,886)	137.2 (13,848)	49.8/108.0/50.5	37.4/43.2/7.2	0.0/7.2/14.4
6 ^α^	53.6 (18,671)	125.3 (16,755)	77.4 (16,786)	16.1/53.7/53.6	10.7/11.9/0.0	0.0/11.9/6.0
7 ^α,γ^	65.6 (48,757)	131.2 (44,196)	109.0 (44,949)	20.5/58.8/46.7 ^α,γ^	16.4/45.3/17.8 ^α, β^	4.1/2.3/11.1
8 ^β^	104.8 (15,268)	129.1 (13,940)	55.6 (14,388)	58.9/71.7/41.7	13.1/14.3/7.0	0.0/7.2/7.0
9	117.7 (26,348)	171.8 (24,454)	130.2 (24,577)	45.5/102.2/16.3 ^α, β^	26.6/40.9/73.2 ^γ^	11.4/4.1/0.0
10 ^α,γ^	61.3 (13,049)	150.6 (11,288)	124.1 (11,283)	30.7/88.6/53.2	7.7/17.7/44.3	15.3/0.0/0.0
11	87.4 (16,012)	94.3 (15,902)	ND	50.0/44.0/ND	25.0/6.3/ND	6.2/6.3/ND
12	44.2 (15,826)	86.8 (14,969)	51.6 (15,501)	19.0/33.4/19.4	19.0/33.4/6.5	0.0/13.4/0.0
13 ^α^	11.1 (9045)	80.9 (8656)	ND	0.0/23.1/ND	11.1/34.7/ND	0.0/11.6/ND
14	34.9 (11,471)	83.2 (9620)	ND	26.2/10.4/ND	8.7/31.2/ND	0.0/10.4/ND
15	86.2 (8116)	14.4 (6946)	ND	37.0/14.4/ND	24.6/0.0/ND	0.0/0.0/ND
16	88.8 (4504)	0.0 (4146)	ND	66.6/0.0/ND	0.0/0.0/ND	0.0/0.0/ND
Overall ^α,γ^	85.4 (359,377)	114.6 (355,060)	112.9 (285,215) **	39.2/58.0/44.9 ^α, β^	21.7/24.5/31.9 ^γ^	5.3/5.4/6.7

ND, not done due to unavailable data about hospital admissions. * Overall incidence was calculated considering all blood culture *Candida* spp, species details reported elsewhere (15, 16). ** Data were calculated removing isolates from hospitals 11 to 16. ^α^ Comparisons of incidence rates between 2019 and 2020 reaching statistical significance (*p* < 0.05). ^β^ Comparisons of incidence rates between 2020 and 2021 reaching statistical significance (*p* < 0.05). ^γ^ Comparisons of incidence rates between 2019 and 2021 reaching statistical significance (*p* < 0.05). Colours indicate the lowest (green), intermediate (pale yellow), and highest incidence rates (red).

**Table 5 jof-08-01228-t005:** Resistant genotypes found in blood cultures and intra-abdominal samples, hospitals, patients involved, and molecular resistance mechanisms.

Species	Genotype	Hospitals Involved	Patients Involved	No. of Isolates and Source		Phenotypic Resistance	FKSp Substitution	ERG11p Substitution
				Blood	Intra -Abdominal			
*C. albicans*	CA-0056 *	H9	1	0	1	Echinocandin	F641L *FKS1* HS1	ND
	CA-0511	H6	2	1	1	Echinocandin	S645P *FKS1* HS1	ND
	CA-1341	H6	1	0	1	Echinocandin	S645P *FKS1* HS1	ND
	CA-1872	H2	1	1	0	Echinocandin	R1361H *FKS1* HS2	ND
	CA-0313 **	H1	1	1	0	Fluconazole	ND	None
	CA-1283	H7	1	1	0	Fluconazole	ND	D115E, K128T, F145L, I471L/I
	CA-1505	H2	1	0	1	Fluconazole	ND	A114S, Y257H
	CA-1674	H1	1	0	1	Fluconazole	ND	None
	CA-1762	H9	1	1	0	Fluconazole	ND	D116E/D, D153E/D
*C. parapsilosis*	CP-540	H9	1	0	1	Echinocandin	None	ND
	CP-451	H2, H4, H9, H15	39	37	2	Fluconazole	ND	Y132F, R398I
	CP-673	H4	2	2	0	Fluconazole	ND	Y132F, R398I
	CP-674	H4	2	2	0	Fluconazole	ND	Y132F, R398I
	CP-675	H10	5	4	1	Fluconazole	ND	G458S
*C. tropicalis*	CT-031 ***	H10	1	0	1	Echinocandin	S654P/S *FKS1* HS1 + V1352I/V, V1404I/V *FKS1*	ND
	CT-081	H6	1	0	1	Fluconazole	ND	None
	CT-313	H7	1	1	0	Fluconazole	ND	F449V

* Genotype also involving susceptible isolates found in hospitals H4, H9, H11, and H15; ** Genotype also involving a susceptible isolate found in hospital H5; *** Genotype also involving susceptible isolates found in hospitals H1 and H10. ND, Not done.

## Data Availability

Not applicable.

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
