# Peer review of "Candida Genotyping of Blood Culture Isolates from Patients Admitted to 16 Hospitals in Madrid: Genotype Spreading during the COVID-19 Pandemic Driven by Fluconazole-Resistant C. parapsilosis"

_jof, 2022, doi:10.3390/jof8111228_

Round 1

Reviewer 1 Report

The Authors describe the results of genotyping Candida spp. isolates collected in 16 hospitals located in the Madrid metropolitan area. 

My comments: 

  • The manuscript should be supplemented with a broader interpretation of the results, incorporate the importance of the resistance to fluconazole;
  • The authors should compare the obtained results with data from other countries/regions/continents. 
  • The results obtained should also be related to data on other antibiotics;
  • The results should be compared to data on other fungi.
  • The manuscript should be editorially improved, including the references.

Author Response

Reviewer 1:

Reviewer comment: The manuscript should be supplemented with a broader interpretation of the results, incorporate the importance of the resistance to fluconazole

Author’s response: We have added the following sentence “Fluconazole resistance is a matter of concern in C. parapsilosis infections given the intrinsic lower susceptibility of the species to echinocandins” (Lines 78-79). We are not sure what the reviewer means by the interpretation of results. We think we already did so; anyway, we have expanded the Discussion section.

Reviewer comment: The authors should compare the obtained results with data from other countries/regions/continents. 

Author’s response: At the best of our knowledge, there are no similar data to compare. However, we have expanded and modified the discussion to emphasize on this matter.

Reviewer comment: The results obtained should also be related to data on other antibiotics. The results should be compared to data on other fungi.

Author’s response: As we stated in the introduction, the data here presented are a sub-analysis of the ones collected in a recently conducted three-year CANDIMAD study (2019 to 2021) study to assess the epidemiology and antifungal susceptibility of a large number of Candida spp isolates from blood cultures and intra-abdominal samples from patients admitted to 16 hospitals located in Madrid, Spain (references 15 and 16). At that time, only data from Candida isolates were collected, therefore we are afraid we will not be able to add additional data on other species or data on antibiotic resistance either.

Reviewer comment: The manuscript should be editorially improved, including the references.

Author’s response: We appreciate the reviewer’ comment. The references have been edited accordingly.

Reviewer 2 Report

The authors analyzed isolates of C. albicansC. parapsilosis and C. tropicalis from 16 hospitals in Madrid that were detected between 2019 and 2021. The isolates were obtained from blood cultures and intra-abdominal samples. A microsatellite genotyping of 1,520 isolates was performed. In addition, the rate of resistance to antifungals was also evaluated, as well as the relationship between resistant genotypes and the origin of clinical samples. The authors showed 83 clusters, which were characterized as intra-hospital or widespread and others as intra-hospital and widespread. An increase in the number of affected patients and clusters was observed during the COVID-19 pandemic, mainly due to the emergence of fluconazole-resistant C. parapsilosis. The manuscript is clear and well written and is relevant as it shows a scenario of infection in an earlier period and shortly after the onset of the COVID-19 pandemic. The methodology is adequate, since microsatellite genotyping is considered discriminative. Therefore, the manuscript may be accepted for publication.

Author Response

We appreciate the reviewer's comments.

Reviewer 3 Report

Díaz-Garcia et al. presents a very interesting and well-designed study about Canddia infections during the COVID-19 pandemic. By genotyping C. albicans, C. parapsilosis, and C. tropicalis isolates from 16 hospitals located in the Madrid metropolitan area, they demonstrated that the number of clusters and patients involved increased during the pandemic. This increase was mainly driven by fluconazole-resistant C. parapsilosis genotypes. The text is well-written, but some alterations are necessary.

-Line 38: The first phrase of the abstract corresponds to methodology. The authors should describe the general objective of the study in the background.

-Lines 39-40: Only the percentage for blood cultures is shown. I suggest removed it or add the percentage of intra-abdominal samples.

-Lines 44-45: Replace “Species-specific microsatellite markers were used to define the genotypes found…..” by ““Species-specific microsatellite markers were used to define the genotypes of Candida isolates found…..”

-Line 83: CANDIMAD seems to be initials of the study’s name. Please, write the complete name.

-Lines 97-98: The proposal of the study should be more detailed and clarified.

-Lines 127-128: Describe the identification of Candida strains used as control.

-Lines 118-148: The item “2.2 Microsatellite genotyping” should be described in more detail. I suggest separate this text in different items with a better description about each methodology used. Also, I suggest including a flowchart to clarify the methods used.

-Figure 2: The size of letter needs to be increased, mainly below the graph.

-Lines 341-343: In the phrase “Some studies demonstrated the rise of candidaemia episodes during the COVID-19 pandemic (9-14) and some of these reported the spread of fluconazole-resistant C. parapsilosis isolates”. Specify the studies that report the spread of C. parapsilosis isolates. Describe these studies in relation to the countries, source of isolates, and others. This part of the discussion should be more explored.

-In general, the discussion is very limited in the previous studies of the authors themselves in Spain. New references must be added and discussed.

Author Response

Reviewer comment: Line 38: The first phrase of the abstract corresponds to methodology. The authors should describe the general objective of the study in the background.

Author’s response: We have restructured the abstract according to the reviewer suggestions (Lines 38-40).

Reviewer comment: -Lines 39-40: Only the percentage for blood cultures is shown. I suggest removed it or add the percentage of intra-abdominal samples.

Author’s response: We have added the percentage of isolates from intra-abdominal samples as well (Line 42).

Reviewer comment: Lines 44-45: Replace “Species-specific microsatellite markers were used to define the genotypes found…..” by ““Species-specific microsatellite markers were used to define the genotypes of Candida isolates found…..”

Author’s response: Done as suggested (Line 63).

Reviewer comment: Line 83: CANDIMAD seems to be initials of the study’s name. Please, write the complete name.

Author’s response: Done as suggested in the Introduction and Abstract (Lines 43-74).

Reviewer comment: Lines 97-98: The proposal of the study should be more detailed and clarified.

Author’s response: We think that the study purpose and objectives are clear in the former version. We gave an idea about the usefulness of genotyping and then offered a summary of the CANDIMAD study and the spreading of fluconazole-resistant C. parapsilosis across the region. Finally, we proposed some unresolved questions that could be elucidated by genotyping the isolates. Maybe the reviewer another question to be added.

Reviewer comment: Lines 127-128: Describe the identification of Candida strains used as control.

Author’s response: Done as suggested (Lines 42 and 74).

Reviewer comment: Lines 118-148: The item “2.2 Microsatellite genotyping” should be described in more detail. I suggest separate this text in different items with a better description about each methodology used. Also, I suggest including a flowchart to clarify the methods used.

Author’s response: The typing methodology used is common to all species, but microsatellite markers used are species-specific. The procedure consists of performing a PCR with species-specific markers, followed by capillary electrophoresis in an automatic sequencer, and results analysis. The entire procedure was already described in the methodology section, and the PCR conditions have been previously published.

Reviewer comment: Figure 2: The size of letter needs to be increased, mainly below the graph.

Author’s response: Done as suggested.

Reviewer comment: Lines 341-343: In the phrase “Some studies demonstrated the rise of candidaemia episodes during the COVID-19 pandemic (9-14) and some of these reported the spread of fluconazole-resistant C. parapsilosis isolates”. Specify the studies that report the spread of C. parapsilosis isolates. Describe these studies in relation to the countries, source of isolates, and others. This part of the discussion should be more explored.

Author’s response: We have rewritten some parts of the discussion, and such comment has been considered (Lines 342-662).

Reviewer comment: In general, the discussion is very limited in the previous studies of the authors themselves in Spain. New references must be added and discussed.

Author’s response: We have rewritten some parts of the discussion, and we have added some information from our previous studies. See the highlighted sections of the discussion.

Reviewer comment: -Lines 39-40: Only the percentage for blood cultures is shown. I suggest removed it or add the percentage of intra-abdominal samples.

Author’s response: We have added the percentage of isolates from intra-abdominal samples as well (Lines 127-129).

Reviewer comment: Lines 44-45: Replace “Species-specific microsatellite markers were used to define the genotypes found…..” by ““Species-specific microsatellite markers were used to define the genotypes of Candida isolates found…..”

Author’s response: Done as suggested well (Lines 127-129).

Reviewer comment: Line 83: CANDIMAD seems to be initials of the study’s name. Please, write the complete name.

Author’s response: Done as suggested in the Introduction and Abstract (Lines 127-129).

Reviewer comment: Lines 97-98: The proposal of the study should be more detailed and clarified.

Author’s response: We think that the study purpose and objectives are clear in the former version. We gave an idea about the usefulness of genotyping and then offered a summary of the CANDIMAD study and the spreading of fluconazole-resistant C. parapsilosis across the region. Finally, we proposed some unresolved questions that could be elucidated by genotyping the isolates. Maybe the reviewer another question to be added.

Reviewer comment: Lines 127-128: Describe the identification of Candida strains used as control.

Author’s response: Added as suggested (Lines 127-129).

Reviewer comment: Lines 118-148: The item “2.2 Microsatellite genotyping” should be described in more detail. I suggest separate this text in different items with a better description about each methodology used. Also, I suggest including a flowchart to clarify the methods used.

Author’s response: The typing methodology used is common to all species, but microsatellite markers used are species-specific. The procedure consists of performing a PCR with species-specific markers, followed by capillary electrophoresis in an automatic sequencer, and results analysis. The entire procedure was already described in the methodology section, and the PCR conditions have been previously published.

Reviewer comment: Figure 2: The size of letter needs to be increased, mainly below the graph.

Author’s response: Done as suggested.

Reviewer comment: Lines 341-343: In the phrase “Some studies demonstrated the rise of candidaemia episodes during the COVID-19 pandemic (9-14) and some of these reported the spread of fluconazole-resistant C. parapsilosis isolates”. Specify the studies that report the spread of C. parapsilosis isolates. Describe these studies in relation to the countries, source of isolates, and others. This part of the discussion should be more explored.

Author’s response: We have rewritten some parts of the discussion, and such comment has been considered.

Reviewer comment: In general, the discussion is very limited in the previous studies of the authors themselves in Spain. New references must be added and discussed.

Author’s response: We have rewritten some parts of the discussion, and we have added some information from our previous studies.

Reviewer 4 Report

The manuscript is a continuation of a series of studies devoted to in-depth assessment of candidiasis agents genetic structure, their resistance to antibiotics, and dissemination between hospitals (Marcos-Zambrano et al., 2015; Escribano et al., 2018; Guinea et al., 2020; Ramos-Martinez et al., 2022, etc). The paper is well structured and written and suits the journal’s scope. It can be recommended for publication after the authors add more details, disclose some abbreviations and reconsider presentation of statistical analysis in certain parts, as proposed below:

Line 157: “C. albicans: 86.7% versus 13.3%” = “86.7% versus 13.3% (C. albicans)” and so on

Lines 158-159: “differences among species” – please specify what you mean by “difference” here

Line 164: I’m not sure I get it right. What do the numbers 272/817 stand for?

Lines 175-176: “the number …. was boosted” – which statistical analysis approach confirms that?

Fig 1: I’m afraid that in the line with 19 patients (CP-451), not all numbers are not clearly seen

Line 183: “latency” is mentioned here but it is not explained what it means and how it was estimated, and is not discussed further in the manuscript

Lines 189-190: please specify whether the statistically significant differences are found between all three values obtained for C. parapsilopsis and make respective designations in Fig. 2.

Line 196: what is ICU?

Line 207: can size of the hospital be indicated in table 1 or elsewhere? Can you support this assumption with statistical analysis?

Table 4: FKSp – is not mentioned in text, while ERG11p is mentioned but not deciphered

Line 324: what are “epidemiologically unrelated patients”

Line 359: please revise wording (“to rule in”)

Line 366: which considerations make you thinking of “the lack of discrimination of the typing procedure”?

Author Response

Reviewer comment: Line 157: “C. albicans: 86.7% versus 13.3%” = “86.7% versus 13.3% (C. albicans)” and so on

Author’s response: Done as suggested (Lines 132-135).

Reviewer comment: Lines 158-159: “differences among species” – please specify what you mean by “difference” here

Author’s response: We mean that when we compared the proportions of clusters found in each of the three species, differences did not reach statistical significance. We have rephrased sentence in lines 134-135, we hope it reads more clearly now.

Reviewer comment: Line 164: I’m not sure I get it right. What do the numbers 272/817 stand for?

Author’s response: 817 stands for the total number of isolates sourcing blood cultures. We have clarified the sentence (Line 151).

Reviewer comment: Lines 175-176: “the number …. was boosted” – which statistical analysis approach confirms that?

Author’s response: Statistical analysis was showed in Figure 2. We have added “Figures 1 and 2” after the sentence. We also added an additional table (Table 4 in the current version) in which we calculated the incidence of candidaemia per hospital, species, and year and some new text (Lines 190-199). We think that such table and Figures 1 and 2 aligns with the observation that during the first few waves of the pandemic the number of clusters increased.

Reviewer comment: Fig 1: I’m afraid that in the line with 19 patients (CP-451), not all numbers are not clearly seen

Author’s response: We appreciate the reviewer comment, the figure has been now modified to make numbers readable.

Reviewer comment: Line 183: “latency” is mentioned here but it is not explained what it means and how it was estimated, and is not discussed further in the manuscript

Author’s response: We meant a time period spanning two time points in which isolates in a cluster were detected. We have added such definition to the manuscript (Lines 165 and 166).

Reviewer comment: Lines 189-190: please specify whether the statistically significant differences are found between all three values obtained for C. parapsilopsis and make respective designations in Fig. 2.

Author’s response: Statistically significant differences in the proportion of patients involved in C. parapsilosis clusters were found between 2019 and 2021, and 2020 and 2021. This has been added to the manuscript (Line 176 and modified Figure 2).

Reviewer comment: Line 196: what is ICU?

Author’s response: It was spelled out as intensive care units (Line 179).

Reviewer comment: Line 207: can size of the hospital be indicated in table 1 or elsewhere? Can you support this assumption with statistical analysis?

Author’s response: We appreciate the comment raised by the reviewer as we also that that information about size of the hospitals were missing. We have added a new table (now Table 4) in which we added the number of admissions per hospital and year; we also calculated the incidence rates of candidaemia (overall and per species) in each hospital and year, and finally compared (and showed) differences in incidence rates. The new table now shows comprehensive data that support the data presented in Figure 2, and the statement that called the reviewer’s attention. Furthermore, we added some text to explain the main observations displayed in Table 4 (Lines 190-199).

Reviewer comment: Table 4: FKSp – is not mentioned in text, while ERG11p is mentioned but not deciphered

Author’s response: The present study complements two previous ones in which detailed data were presented, as stated in the introduction. Therefore, we focused here on relevant information under the perspective of clonal spreading of resistant isolates. Since echinocandin-resistant isolates did not spread, we think that there is no need to emphasize on that. The only echinocandin-resistant genotype involving two patients was already discussed in the former draft.

Reviewer comment: Line 324: what are “epidemiologically unrelated patients”

Author’s response: We meant “patients admitted to different hospitals”. It has been clarified in the current revised draft (Line 70 and Lines 329 and 330).

Reviewer comment: Line 359: please revise wording (“to rule in”)

Author’s response: The reviewer is right, we meant that we cannot exclude the environmental niche. We have changed it into “rule out”.

Reviewer comment: Line 366: which considerations make you thinking of “the lack of discrimination of the typing procedure”?

Author’s response: Microsatellites are valuable genotyping tools. However, the presence of identical genotypes without a definite epidemiological relationship among patients should be interpreted with caution; homoplasy events may lead to identical genotypes as per microsatellites may have differences in other regions of the genome. Therefore, whole genome sequencing is a promising tool for future investigations, and we are currently working on it.

Round 2

Reviewer 1 Report

The manuscript has been improved. Most of the comments have been taken into account. However, editorial errors need to be corrected, e.g., the species names of microorganisms should be written in italics (this is especially evident in the references), and dots need to be added at the end of some sentences.